# Physical fitness and stroke performance in healthy tennis players with different competition levels: A systematic review and meta-analysis

**Johanna Lambrich** *, **Thomas Muehlbauer**

Division of Movement and Training Sciences/Biomechanics of Sport, University of Duisburg-Essen, Essen, Germany

* johanna.lambrich@uni-due.de

**Data Availability Statement:** All relevant data are within the paper and its Supporting information files.

## Abstract

Differences in variables of physical fitness and stroke performance by competition level (i.e., elite vs. sub-elite players) have not been systematically investigated yet. Thus, the objective of the systematic review with meta-analysis was to characterize and quantify competition-level dependent differences in physical fitness and stroke performance in healthy tennis players. A systematic literature search was conducted in the databases PubMed, Web of Science, and SportDiscus from their inception date till May 2022. Studies were included if they investigated healthy tennis players and reported at least one measure of physical fitness (e.g., lower extremity muscle power, endurance, agility, speed) or stroke performance (e.g., stroke velocity). Weighted standardized mean differences ($SMD$) were calculated and reported according to their magnitude. The search identified a total of $N = 12,714$ records, 16 of which met the inclusion criteria. Competition-level dependent differences in physical fitness and stroke performance were investigated by 11 and 10 studies, respectively. For physical fitness, moderate (lower extremity muscle power: $SMD = 0.53$; endurance: $SMD = 0.59$; agility: $SMD = 0.54$) and small (speed: $SMD = 0.35$) effects were detected; all in favour of elite tennis players. However, sub-group analyses revealed an influence of players' age showing higher $SMD$-values for adult than for young players. Further, a large effect ($SMD = 1.00$) was observed for stroke performance again in favour of elite tennis players. Lastly, a larger but not significantly different association between physical fitness and stroke performance was observed for elite ($r = 0.562$) compared to sub-elite ($r = 0.372$) tennis players. This systematic review and meta-analysis revealed better physical fitness and stroke performances in healthy elite compared to sub-elite tennis players. The greatest differences by competition level were shown in measures of lower extremity muscle power, endurance, and agility. Thus, training programs for sub-elite tennis players should place a special focus on these physical components.

**Funding:** The authors received no specific funding for this work.

**Competing interests:** The authors have declared that no competing interests exist.

## Introduction

In tennis, various components of physical fitness play an important role. On the one hand, the average duration for scoring a point is less than 3–10 s [1, 2] with sprints of 8–15 m (i.e., speed) and 3–4 directional changes (i.e., agility) taking place during rallies [3–5]. On the other hand, a match can last up to five hours, which shows the importance of tennis-specific endurance [1, 2]. Further, different components of strength (e.g., lower extremity muscle power) are important to perform tennis-specific footwork and explosive strokes during rallies [2, 6]. Therefore, an optimal development of the previously described physical fitness components seems to be an important prerequisite for a good stroke performance (e.g., high ball speed) and thus for success in a tennis match [7].

Indeed, in studies using a between-subject-design, both better physical fitness scores [6, 8, 9] and higher stroke performance [10–12] have been reported for elite compared to sub-elite tennis players. In addition, studies that used a within-subject design showed that higher stroke performance was associated with better physical fitness values [4, 13]. Despite these findings, a systematic review of studies on differences in physical fitness and stroke performance depending on competition level is still lacking. Specifically, the aggregation and quantification of performance differences is important in order to make deductions for the design of training programs in tennis. For example, physical fitness components with the greatest discrepancies between elite and to sub-elite tennis players can be identified so that these can be particularly addressed for the latter one in the context of training.

Therefore, the aim of the present systematic review and meta-analysis was to aggregate and quantify differences in physical fitness and stroke performance in healthy tennis players by competition level. With reference to the relevant literature that used a between-subject-design [10, 14, 15], we expected better physical fitness and stroke performance in and healthy elite compared to sub-elite tennis players. Further and considering previous findings from studies using a within-subject design [4, 13], we assumed larger correlations between physical fitness and stroke performance in elite than in sub-elite players.

## Methods

### Search strategy

To identify relevant literature for this review, a systematic literature search was conducted in the electronic databases PubMed, Web of Science, and SPORTDiscus. The following Boolean search term was used: (tennis AND ((performance level OR competition level OR elite OR expert OR high performance OR non-expert OR sub-elite OR amateur NOR Novice NOR beginner) OR (stroke OR physical fitness OR fitness characteristics OR agility OR endurance OR speed OR muscle power OR lower extremity NOT upper extremity) OR (forehand OR backhand OR serve OR volley OR overhead)) NOT table). The search covered the period from the first publication to May 2022. The literature search was limited to English language, human species, and to full text original articles. In addition, reference lists of the included studies and relevant reviews were searched for additional articles.

After removing duplicates, the titles and abstracts of all records were screened independently by both authors for eligibility according to the inclusion and exclusion criteria as stated in Table 1. The full-text version of an article was retrieved and screened for eligibility if the information provided in the title and abstract was insufficient. Afterwards, full-text versions of all potentially relevant studies were obtained and assessed for inclusion independently by both authors. Disagreement was resolved by discussion and consensus. The process of literature

**Table 1. Overview of the inclusion and exclusion criteria.**

| Category | Inclusion criteria | Exclusion criteria |
|---|---|---|
| Population | healthy tennis players (12–32 years) | injured tennis players; no tennis players; beginner tennis players |
| Measurement | fitness tests; sport-specific test | cognitive test only |
| Outcome | at least one parameter of physical fitness or stroke performance | data did not allow to calculate effect size |
| Study design | cross-sectional study; longitudinal study | intervention study not reporting pretest data; review; meta-analysis |

search, study selection, and reasons for exclusion of records are documented in Fig 1 by using the PRISMA flow chart [16].

## Study selection criteria

The applied predefined criteria for selection are presented in Table 1. To be eligible for inclusion, studies had to meet the following criteria: a) tennis players were healthy and aged between 12 and 32 years, b) a physical fitness or sport-specific test was performed, c) at least one physical fitness or stroke performance outcome was reported, d) a cross-sectional or longitudinal study design was used. Studies were excluded if: a) they investigated injured, beginner or no tennis players (i.e., table tennis players), b) only a cognitive test was applied, c) the provided data did not allow the calculation of effect size and the corresponding author did not reply to our inquiry, and d) an intervention was applied but no pretest data were reported.

## Study coding

The included studies were coded due to the following criteria: authors and publication year, number of tennis players by sex, age, competition level, stroke/physical performance test, and outcomes. Because of differences in the terminology used to define the competition level, we classified the groups "elite" and "sub-elite" tennis players based on the information provided in each article. In this regard, international, professional, high-performance, higher-ranked, world class, division I, and Davis cup players were rated as "elite" players. In contrast, intermediate, competitive, advanced, club, lower-ranked, regional, skilled, division II, and junior players were evaluated as "sub-elite" players.

Physical fitness is defined as a set of attributes that people have or achieve and that can be measured with specific tests [17]. It was classified into the following categories: muscle power of the lower extremities, endurance, agility, and speed. Further, stroke performance was characterized through sport-specific assessment methods (e.g., Dutch Technical-Tactical Test [18] or serve tests). Since some studies reported several variables within one outcome category, we preferred the most frequently reported measure for each category to reduce heterogeneity between studies (Table 2). Regarding lower extremity muscle power, the highest priority was given to countermovement jump height, while the maximal level achieved during a tennis specific endurance test was used with reference to endurance. In terms of agility, the highest relevance was granted to the time for the spider test, while time for the 10-m sprint test was defined as most representative for speed. Finally, mean stroke velocity was defined as most representative for stroke performance. If studies reported another measure as proxy for the aforementioned categories, an alternative outcome was used (Table 2).

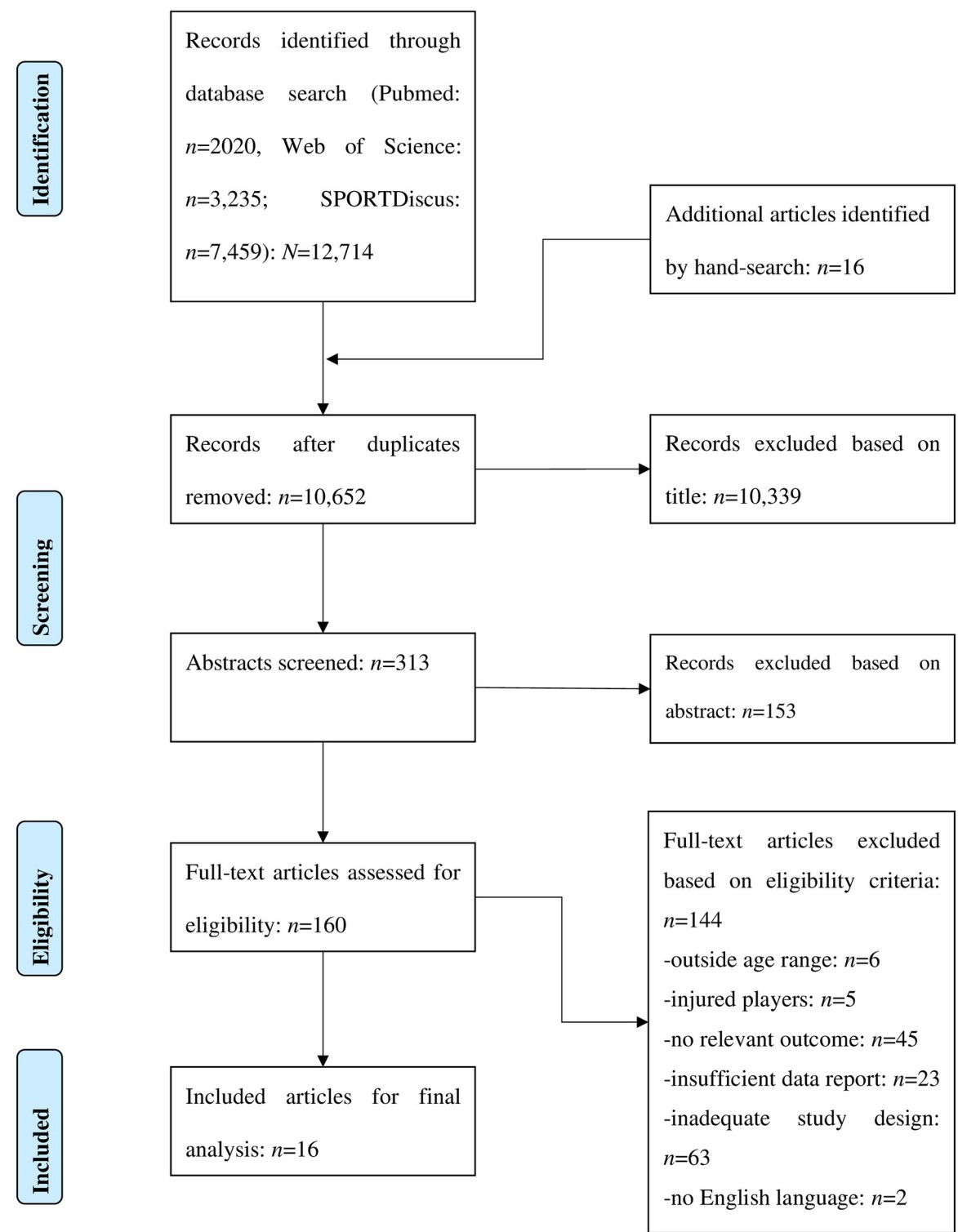

**Fig 1. PRISMA flow chart illustrating the different phases of literature search, study selection, and reasons for exclusion of records.**

**Table 2. Overview of the preferred and alternative outcome by category.**

| Category | Preferred outcome | Alternative outcome |
|---|---|---|
| Muscle power | countermovement jump height in cm ($n = 5$) | vertical jump height in cm ($n = 2$) leg stiffness in kN/m ($n = 2$) |
| Endurance | final stage number ($n = 3$) | $VO_2$max in ml/min/kg ($n = 2$) |
| Agility | spider test time in seconds ($n = 2$) | Illinois agility run time seconds ($n = 1$) |
| Speed | 10-m sprint time in seconds ($n = 2$) | 5-m sprint time seconds ($n = 1$) 40-m-sprint time seconds ($n = 1$) |
| Stroke performance | mean stroke velocity in km/h ($n = 6$) | maximal stroke velocity in km/h ($n = 4$) |

Note. The figure in brackets indicates the number of studies that made use of the test.

### Assessment of study quality

To assess the quality of the included studies, we used the appraisal tool for cross-sectional studies [19]. The tool consists of 20 questions that must be answered with "yes", "no", or "do not know". Seven questions (1, 4, 10, 11, 12, 16, 18) refer to the quality of reporting and further seven questions (2, 3, 5, 8, 17, 19, 20) to the study design. Another six questions (6, 7, 9, 13, 14, 15) relate to a possible risk of bias. Three questions (7, 13, 14) that ask for potential non-responders were excluded from the analysis as this criterion was not applicable for the vast majority of included studies. Quality assessment was independently performed by both authors and disagreement was resolved by discussion and consensus. A score above the median was indicative for adequate study quality.

### Statistical analysis

The standardized mean difference (*SMD*) was used to examine differences due to competition level (i.e., elite vs. sub-elite tennis players) including all player (12–32 years) and for young (<18 years) compared to adult (≥18 years) players [20]. The formula to calculate the *SMD* is: SMD = difference of means / pooled standard deviation. For each outcome measure, a weighted mean *SMD* was computed using Review Manager version 5.4.1. A positive *SMD* indicates better physical fitness/stroke performance in elite compared to sub-elite tennis players. According to Cohen [21], the *SMD* can be interpreted as follows: small ($0 \leq 0.49$), moderate ($0.50 \leq 0.79$), or large ($\geq 0.80$) effect. To quantify the heterogeneity between studies, the $I^2$ and $Chi^2$ statistics were applied. In accordance to Deeks et al. [22], heterogeneity was reported as trivial ($0 \leq 40\%$), moderate ($30 \leq 60\%$), substantial ($50 \leq 90\%$), or considerable ($75 \leq 100\%$). Further, associations between physical fitness and stroke performance were assessed using the Pearson product-moment correlation coefficient. Associations were reported by the correlation coefficient (*r*-value), the level of significance (*p*-value), and the amount of variance explained ($R^2$-value). Values of $0 \leq r \leq 0.49$ indicate a small, $0.50 \leq r \leq 0.69$ a moderate, and $0.70 \leq r \leq 0.99$ a high correlation [21]. In addition, we assessed the significance of the difference between the *r*-values obtained for elite versus sub-elite tennis players using the Fisher r-to-z transformation [23]. The corresponding formula is: $z' = 0.5[\ln(1+r)—\ln(1-r)]$. The significance level was set at $\alpha = 5\%$. All analyses were performed using the Statistical Package for Social Sciences (SPSS) version 27.0.

## Results

### Study selection

Fig 1 illustrates the individual stages of the systematic literature search and the process of study selection as well as the reasons for exclusion of records. Initially, the search revealed 12,714 records for appraisal. Additionally, 16 records were identified through other sources.

After removal of duplicates and screening of titles and abstracts, 160 full-text articles were assessed for eligibility. Of these, 144 articles were excluded for the following reasons: players in 6 studies were outside the age range of 12–32 years, 5 articles involved injured players, 45 records did not conduct a fitness/sport-specific test, 23 articles did not provide sufficient information on outcome measures used and this information could not be retrieved from the corresponding author, 63 records did not use an adequate study design, and two records did not use English language. The remaining 16 studies met all of our inclusion criteria and were used for the final analysis, including four which investigated multiple age cohorts.

## Study characteristics

Table 3 illustrates the main characteristics of the included studies. Comparisons between elite and sub-elite tennis players were conducted in eleven studies for variables of physical fitness [6, 8–10, 14, 15, 24–28] and in ten studies for measures of stroke performance [10–12, 14, 15, 18, 24, 26, 29, 30]. In total, 1,794 players (i.e., 600 elite and 1,194 sub-elite) were investigated in the included studies. Nine studies contained male players only, three studies examined female players and the remaining four studies included players of both sexes. Seven studies investigated junior players in the age between 12 and 18 years. Four studies examined adults, and the age in the remaining five studies ranged from 15 to 26 years. Regarding physical fitness, nine studies assessed muscle power of the lower extremities, four studies measured speed, three studies investigated agility, and five studies evaluated endurance. Further, seven studies used the serve and another four studies used the forehand and backhand during assessment of stroke performance.

## Study quality

The assessment of study quality revealed that 16 out of 16 studies fulfilled $\geq 4$ out of 7 criteria regarding the quality of reporting, 16 out of 16 studies fulfilled $\geq 4$ out of 7 criteria addressing the study design, 16 out of 16 studies fulfilled $\geq 2$ out of 3 criteria concerning risk of bias (S1 Table). In sum, all included studies met the criteria for study quality above average.

## Physical fitness differences by competition level

The comparisons of physical fitness between elite and sub-elite tennis players are shown in Figs 2–5. Weighted mean *SMD* amounted to 0.53 for outcomes of lower extremity muscle power ($Chi^2$ = 49.13, *df* = 21, *p* = .0005, 9 studies, 22 comparisons), 0.59 for variables of endurance ($Chi^2$ = 9.69, *df* = 7, *p* = .21, 5 studies, 8 comparisons) and 0.54 for measures of agility ($Chi^2$ = 10.13, *df* = 10, *p* = .43, 3 studies, 11 comparisons) indicating moderate effects in favor of elite players (Figs 2–4). As shown in Fig 5, a small effect in favor of elite players was found for parameters of speed as *SMD* amounted to 0.35 ($Chi^2$ = 30.06, *df* = 17, *p* = .03, 4 studies, 18 comparisons). Heterogeneity between studies was trivial for measures of endurance ($I^2$ = 28%) and agility ($I^2$ = 1%), moderate for speed ($I^2$ = 43%), and substantial for variables of lower extremity muscle power ($I^2$ = 57%). Further, the age-specific sub-analysis revealed that *SMD*-values for lower extremity muscle power, endurance, and agility were large in adult but small to moderate in young players (Table 4) that is indicative of a moderating effect of age. For speed, no age-related differences were detected.

## Stroke performance differences by competition level

The comparisons of stroke performance (i.e., stroke velocity) between elite and sub-elite tennis players is displayed in Fig 6. Weighted mean *SMD* amounted to 1.00 ($Chi^2$ = 25.81, *df* = 17, *p* =

**Table 3. Overview of the included studies comparing physical fitness and stroke performance between players with different competition level.**

| Reference | No. of players | | | Age [years (range or mean ± SD)] | Competition level | Stroke performance test; outcome | Physical performance test; outcome |
|---|---|---|---|---|---|---|---|
| | All | M | F | | | | |
| Elliott et al. [8] | 143 | 78 | 65 | 13–15 | high-performance vs. competitive | | *Muscle power*: Vertical jump [cm] |
| | | | | | | | *Speed*: 40-m sprint [s] |
| | | | | | | | *Agility*: Illinois agility run [s] |
| Girard et al. [24] | 32 | 32 | 0 | 21.5 ± 3.8 | elite vs. intermediate | *Serve test*: stroke velocity [km/h] | *Muscle power*: Leg stiffness [kN/m] |
| Landlinger et al. [11] | 13 | 13 | 0 | 15–26 | elite vs. high-performance | *Forehand and backhand test*: maximal stroke velocity [km/h] | |
| Martin et al. [29] | 18 | 18 | 0 | 18–32 | elite vs. advanced | *Serve test*: stroke velocity [km/h] | |
| Baiget et al. [25] | 38 | 38 | 0 | 16–20 | international vs. national | | *Endurance*: final stage [no.] |
| Kramer et al. [9] | 87 | 87 | 0 | 12–13 | higher- vs. lower-ranked | | *Muscle power*: CMJ [cm] |
| | 66 | 0 | 66 | 12–13 | | | *Speed*: 10-m sprint [s] |
| | 79 | 79 | 0 | 13–14 | | | *Agility*: Spider test [s] |
| | 55 | 0 | 55 | 13–14 | | | |
| | 54 | 54 | 0 | 14–15 | | | |
| | 37 | 0 | 37 | 14–15 | | | |
| Ulbricht et al. [14] | 255 | 255 | 0 | 12–14 | national vs. regional | *Serve test*: mean stroke velocity [km/h] | *Muscle power*: CMJ [cm] |
| | 165 | 165 | 0 | 14–16 | | | *Speed*: 10-m sprint [s] |
| | | | | | | | *Endurance*: |
| | 177 | 0 | 177 | 12–14 | | | Final stage [no.] |
| | 97 | 0 | 97 | 14–16 | | | |
| Brechbuhl et al. [26] | 27 | 0 | 27 | 16.7 ± 3.1 | elite vs. junior | *Forehand and backhand test*: mean stroke velocity [km/h] | *Endurance*:Final stage [no.] |
| Sögüt et al. [30] | 17 | 0 | 17 | 12–14 | elite vs. club | *Serve test*: stroke velocity [km/h] | |
| Mecheri et al. [27] | 16 | 16 | 0 | 18–31 | world class vs. skilled | | *Muscle power*: Leg stiffness [kN/m] |
| Özkatar Kaya et al. [6] | 20 | 20 | 0 | 19–25 | division I vs. division II | | *Muscle power*: Vertical jump [cm] |
| | | | | | | | *Agility*: Spider test [s] |
| | | | | | | | *Endurance*: VO$_2$max [ml/min/kg] |
| Fett et al. [15] | 178 | 131 | 47 | 14–17 | Davis cup vs. regional | *Serve test*: mean stroke velocity [km/h] | *Muscle power*: CMJ [cm] |
| | | | | | | | *Speed*: 10-m sprint [s] |
| | | | | | | | *Endurance*: VO$_2$max [ml/min/kg] |
| Filipcic et al. [12] | 16 | 16 | 0 | 13–22 | professional vs. junior | *Stroke performance (serve, return, volley, forehand and backhand)*: mean stroke velocity [km/h] | |
| Kolman et al. [31] | 29 | 29 | 0 | 13.4 ± 0.51 | elite vs. competitive | *Dutch Technical-Tactical Tennis Test*: mean stroke velocity [km/h] | |
| Kramer et al. [28] | 80 | 0 | 80 | 12–13 | elite vs. sub-elite | | *Muscle power*: CMJ [cm] |
| | 52 | 0 | 52 | 13–14 | | | *Speed*: 5-m sprint [s] |
| | 28 | 0 | 28 | 14–15 | | | |
| Sanchez-Pay et al. [10] | 15 | 15 | 0 | 19.66 ± 1.63 | professional vs. national | *Serve test*: mean stroke velocity [km/h] | *Muscle power*: CMJ [cm] |

Note. CMJ: countermovement jump; F: female; M: male; SD: standard deviation.

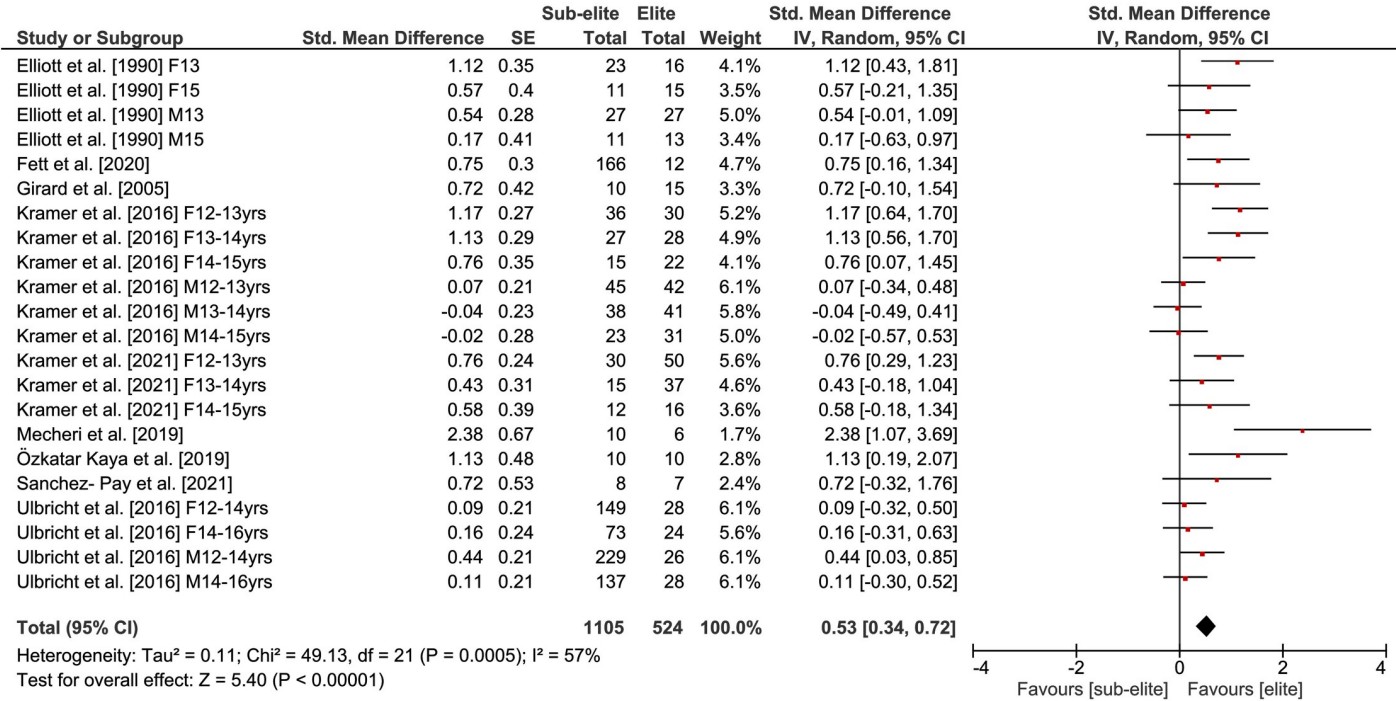

**Fig 2. Differences in measures of lower extremity muscle power by competition level (i.e., elite vs. sub-elite tennis players).** *CI* = confidence interval, *df* = degrees of freedom, *SE* = standard error, *IV* = inverse variance.

.08, 10 studies, 18 comparisons) indicating a large effect in favor of elite players. Heterogeneity between studies was trivial ($I^2$ = 34%). The additionally performed age-specific sub-analysis showed large *SMD*-values for both the young and the adult players (Table 4), indicating no moderating role of age.

## Associations between physical fitness and stroke performance by competition level

Fig 7 illustrates the correlations of physical fitness (i.e., muscle power of the lower extremities) and stroke performance (i.e., stroke velocity) by competition level. We observed a moderate

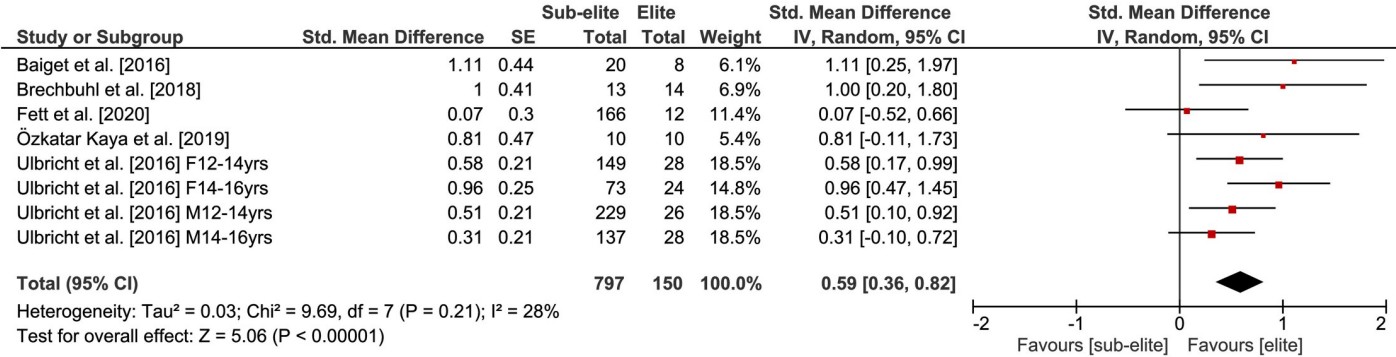

**Fig 3. Differences in measures of endurance by competition level (i.e., elite vs. sub-elite tennis players).** *CI* = confidence interval, *df* = degrees of freedom, *SE* = standard error, *IV* = inverse variance.

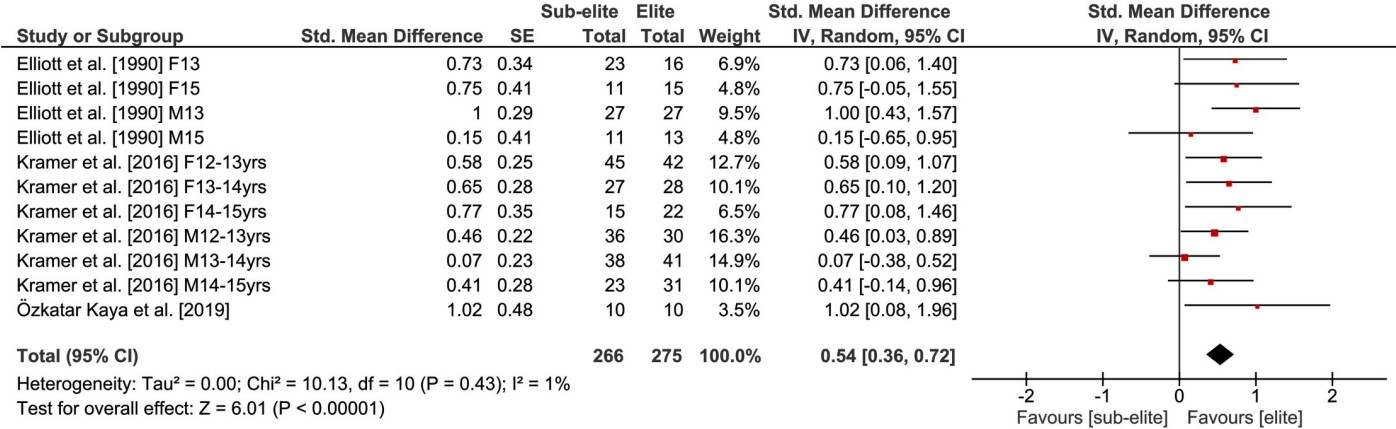

**Fig 4. Differences in measures of agility by competition level (i.e., elite vs. sub-elite tennis players).** *CI* = confidence interval, *df* = degrees of freedom, *SE* = standard error, *IV* = inverse variance.

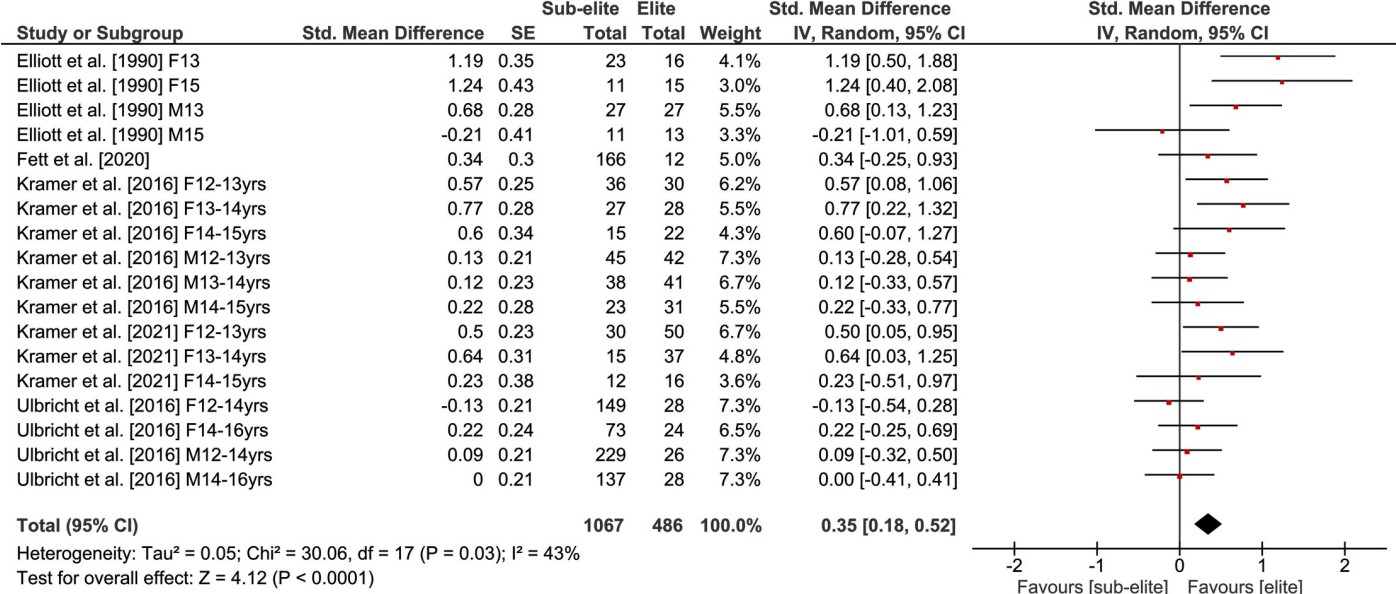

**Fig 5. Differences in measures of speed by competition level (i.e., elite vs. sub-elite tennis players).** *CI* = confidence interval, *df* = degrees of freedom, *SE* = standard error, *IV* = inverse variance.

**Table 4. Differences in measures of physical fitness and stroke performance between players with different competition level (i.e., elite vs. sub-elite tennis players) by age group.**

| Measure | Young players (<18 years) | Adult players (≥18 years) | All players (12–32 years) |
| --- | --- | --- | --- |
| Muscle power | 0.46 | 1.12 | 0.53 |
| Endurance | 0.50 | 0.98 | 0.59 |
| Agility | 0.52 | 1.02 | 0.54 |
| Speed | 0.35 | – | 0.35 |
| Stroke performance | 0.95 | 1.15 | 1.00 |

Note. Data are presented as standardized mean difference.

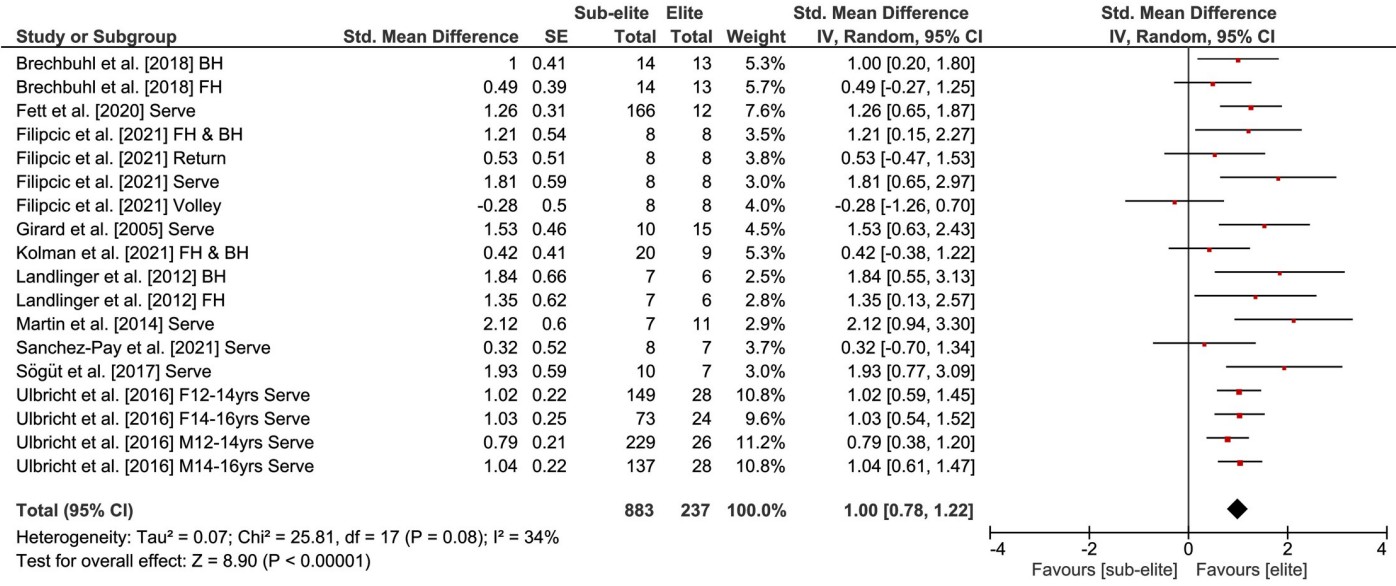

| Study or Subgroup | Std. Mean Difference | SE | Sub-elite Total | Elite Total | Weight | Std. Mean Difference IV, Random, 95% CI |
|---|---|---|---|---|---|---|
| Brechbuhl et al. [2018] BH | 1 | 0.41 | 14 | 13 | 5.3% | 1.00 [0.20, 1.80] |
| Brechbuhl et al. [2018] FH | 0.49 | 0.39 | 14 | 13 | 5.7% | 0.49 [-0.27, 1.25] |
| Fett et al. [2020] Serve | 1.26 | 0.31 | 166 | 12 | 7.6% | 1.26 [0.65, 1.87] |
| Filipcic et al. [2021] FH & BH | 1.21 | 0.54 | 8 | 8 | 3.5% | 1.21 [0.15, 2.27] |
| Filipcic et al. [2021] Return | 0.53 | 0.51 | 8 | 8 | 3.8% | 0.53 [-0.47, 1.53] |
| Filipcic et al. [2021] Serve | 1.81 | 0.59 | 8 | 8 | 3.0% | 1.81 [0.65, 2.97] |
| Filipcic et al. [2021] Volley | -0.28 | 0.5 | 8 | 8 | 4.0% | -0.28 [-1.26, 0.70] |
| Girard et al. [2005] Serve | 1.53 | 0.46 | 10 | 15 | 4.5% | 1.53 [0.63, 2.43] |
| Kolman et al. [2021] FH & BH | 0.42 | 0.41 | 20 | 9 | 5.3% | 0.42 [-0.38, 1.22] |
| Landlinger et al. [2012] BH | 1.84 | 0.66 | 7 | 6 | 2.5% | 1.84 [0.55, 3.13] |
| Landlinger et al. [2012] FH | 1.35 | 0.62 | 7 | 6 | 2.8% | 1.35 [0.13, 2.57] |
| Martin et al. [2014] Serve | 2.12 | 0.6 | 7 | 11 | 2.9% | 2.12 [0.94, 3.30] |
| Sanchez-Pay et al. [2021] Serve | 0.32 | 0.52 | 8 | 7 | 3.7% | 0.32 [-0.70, 1.34] |
| Sögüt et al. [2017] Serve | 1.93 | 0.59 | 10 | 7 | 3.0% | 1.93 [0.77, 3.09] |
| Ulbricht et al. [2016] F12-14yrs Serve | 1.02 | 0.22 | 149 | 28 | 10.8% | 1.02 [0.59, 1.45] |
| Ulbricht et al. [2016] F14-16yrs Serve | 1.03 | 0.25 | 73 | 24 | 9.6% | 1.03 [0.54, 1.52] |
| Ulbricht et al. [2016] M12-14yrs Serve | 0.79 | 0.21 | 229 | 26 | 11.2% | 0.79 [0.38, 1.20] |
| Ulbricht et al. [2016] M14-16yrs Serve | 1.04 | 0.22 | 137 | 28 | 10.8% | 1.04 [0.61, 1.47] |
| **Total (95% CI)** | | | **883** | **237** | **100.0%** | **1.00 [0.78, 1.22]** |

Heterogeneity: Tau² = 0.07; Chi² = 25.81, df = 17 (P = 0.08); I² = 34%
Test for overall effect: Z = 8.90 (P < 0.00001)

**Fig 6. Differences in measures of stroke performance (i.e., stroke velocity) by competition level (i.e., elite vs. sub-elite tennis players).** *CI* = confidence interval, *df* = degrees of freedom, *SE* = standard error, *IV* = inverse variance.

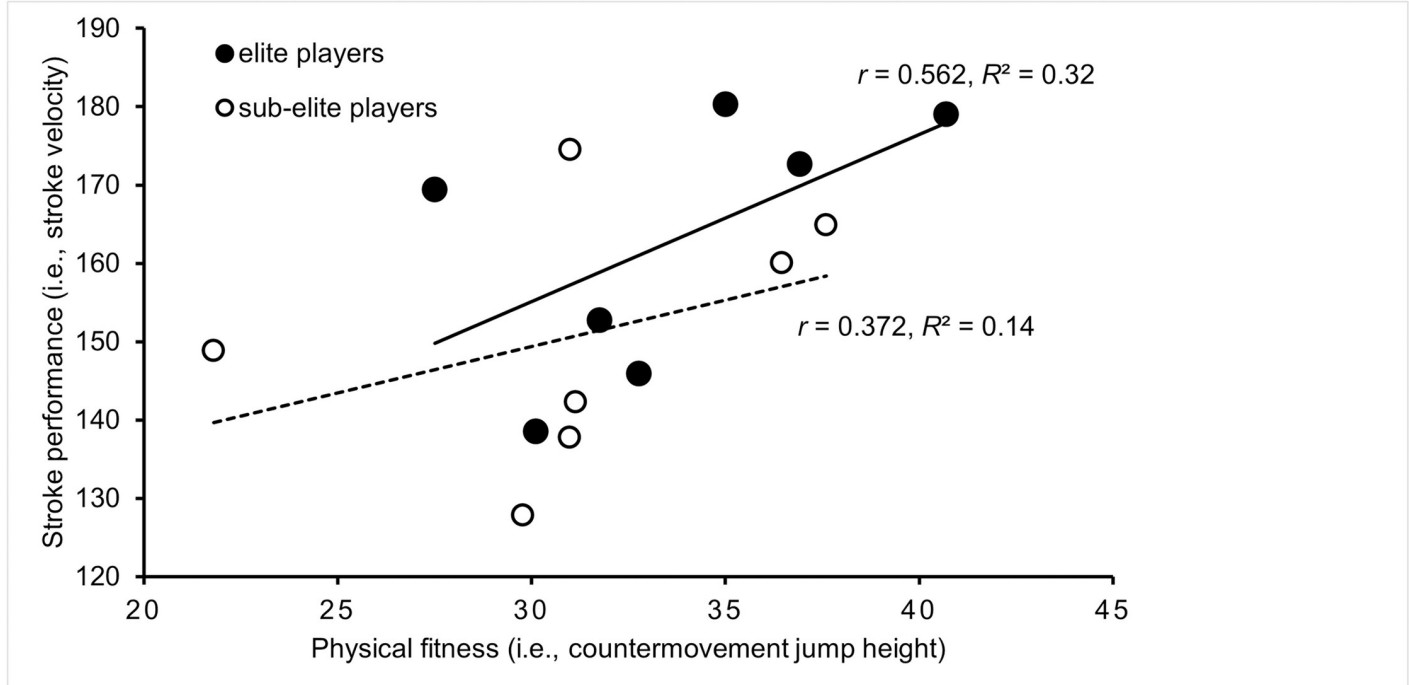

**Fig 7. Correlations between physical fitness (i.e., lower extremity muscle power) and stroke performance (i.e., stroke velocity) by competition level.** Filled circles and the solid regression line indicate elite tennis players and unfilled circles and the dotted regression line indicate sub-elite tennis players. *r* = Pearson's correlation coefficients, $R^2$ = coefficient of determination.

correlation in elite players ($r = 0.562$, $p = 0.190$, $R^2 = 0.32$) and a low correlation in sub-elite players ($r = 0.372$, $p = 0.411$, $R^2 = 0.14$). The comparison of $r$-values between competition levels did not reach the level of significance ($z' = 0.346$, $p = 0.364$).

## Discussion

This systematic review with meta-analysis characterized, aggregated, and quantified competition-level dependent differences (i.e., elite vs. sub-elite) in physical fitness and stroke performance in healthy young tennis players. The main findings can be summarized as follows. First, moderate (i.e., lower extremity muscle power, endurance, agility) and small (i.e., speed) effects, all in favour of elite players, were found for variables of physical fitness. Second, for measures of stroke performance (i.e., stroke velocity) a large effect was detected, again in favour of elite players. Third, correlations between physical fitness (i.e., lower extremity muscle power) and stroke performance (i.e., stroke velocity) were moderate in elite but low in sub-elite tennis players. However, they did not significantly differ by competition level.

### Differences in physical fitness by competition level

Our analyses revealed better physical fitness of elite compared to sub-elite tennis players in terms of lower extremity muscle power, endurance, agility, and speed. These findings support our first hypothesis and are in line with the notion that long-term training leads to improvements of physical fitness components [9, 32]. However, the magnitude of the detected differences varied depending on the dimension of physical fitness. Specifically, a small effect size for speed but moderate effect sizes for lower extremity muscle power, endurance, and agility were found. One possible reason for these differences in magnitude could be the nature of the underlying physical fitness component. Speed is primarily an informationally determined component for which the processes of receiving, processing, and transmitting information are particularly important [33]. These processes are largely genetically determined [34], and thus the potential for training-induced adaptations is relatively low, resulting in small-sized differences as a function of competition level. Although agility is also an informationally determined fitness component, it involves complex requirements [35, 36]. For example, in the Spider test (i.e., agility) contrary to the 10-m linear sprint (i.e., speed), it is necessary i) to change direction (i.e., to the left and to the right), ii) to complete different running paths (i.e., straight and diagonal), and iii) to realize several successive tasks (i.e., pick-up, carry, and put-down a tennis ball). Thus, cognitive and perceptual processes are required in addition to the reception, processing, and transmission of information. These processes are mainly developed through years of training [37, 38], which explains the moderate-sized differences in favour of the elite tennis players. Another factor is that during a tennis match, linear sprints are less common compared to agility requirements such as combining different running styles and directions [39]. Thus, agility seems to be more important than speed, but requires several years of training due to their complex nature [37], which also explains the moderate-sized differences in favour of the elite tennis players.

In contrast, endurance and lower extremity muscle power are more energetically determined fitness components for which processes like energy supply and transmission are particularly important [1, 34]. Thus, endurance and muscle power of the lower extremity can be trained comparatively well and thus have a high potential for adaptation [34]. This could explain the larger differences for these two fitness components between elite and sub-elite tennis players. From a practical perspective, the largest differences between these two groups were detected for measures of endurance, lower extremity muscle power, and agility, which

indicates that a particular focus in the training of sub-elite tennis players should be placed on these physical fitness components.

In addition, the size of the *SMD*-values differed by age group for measures of lower extremity muscle power, endurance, and agility. Specifically, differences between elite and sub-elite players in these physical fitness components were greater for adult than for young players. This implies that with increasing age it becomes more important to train lower extremity muscle power, endurance, and agility in order to perform successful. In this regard, Kurtz et al. reported a significant correlation between national ranking and agility in adult tennis players [4]. In contrast, no significant correlations between national ranking and lower extremity muscle power were detected for young players [14, 40].

### Differences in stroke performance by competition level

In addition, the analyses yielded better stroke performance of elite compared to sub-elite tennis players with respect to stroke velocity. This result additionally supports our first hypothesis and shows that continuous training is associated with improvements in sport-specific performance [40]. The detected difference between elite and sub-elite tennis players can be classified as large-sized. To show a high sport-specific performance level, technical and tactical aspects are required in addition to physical fitness components, i.e., motor skills and cognitive processes that can only be acquired through years of practice/learning and thus have a high potential for adaptation [41]. This most likely explains the large-sized effect for stroke performance as a function of competition level. Further, the size of the *SMD*-values did not differ by age group and was large in young and adult players as well. This suggests that it is significant to practice tennis specific skills already at a young age. In fact, it is recommended in the guidelines of the German Tennis Confederation to practice stroke techniques especially in young players (i.e., from under 10 to under 14 years) [42].

### Difference in correlations between physical fitness and stroke performance by competition level

Due to limited data available, the calculation of correlations between physical fitness and stroke performance was only possible for parameters of lower extremity muscle power and stroke velocity. The result showed that the greater the jump height, the faster the stroke velocity. Although the observed correlations were not significant, they varied in magnitude depending on the competition level. More specifically and consistent with our second hypothesis, the *r*-value was moderate for elite but low for sub-elite tennis players. This suggests that the level of lower extremity muscle power explains a greater proportion of variance with respect to stroke performance in elite than in sub-elite tennis players [24]. From a practitioner's perspective, it can be deduced that training-induced gains in lower extremity muscle power can be transferred to improvements in stroke speed, at least to some degree and that elite tennis players in particular benefit from this.

### Limitations

There are a few limitations with this systematic review and meta-analysis that need to be addressed. First, the terminology used to distinguish between elite and sub-elite tennis players included a variety of terms and may therefore have had an impact on the assignment. Second, due to limited data a calculation of correlations was only possible between stroke performance and muscle power of the lower extremity but not for other physical fitness parameters (i.e., endurance, agility, speed). Third, only studies with tennis players in the age range of 12–32 years were identified, thus no statement can be made about younger or older players.

## Conclusions

The present systematic review with meta-analysis aggregated and quantified competition-level dependent differences in physical fitness and stroke performance in tennis players. We found better physical fitness values (i.e., lower extremity muscle power, endurance, agility, speed) and stroke performance levels (i.e., stroke velocity) in young healthy elite compared to sub-elite players. The largest discrepancies in physical fitness were observed for lower extremity muscle power, endurance, and agility, so that these components should be especially trained in sub-elite tennis players. In addition, low and moderate correlations were found between physical fitness (i.e., lower extremity muscle power) and stroke performance (i.e., stroke velocity) for sub-elite and elite tennis players, respectively. This indicates that gains made in lower extremity muscle power after strength training may be associated with improvements in stroke performance (e.g., stroke velocity) and should be investigated in future studies.

## Supporting information

**S1 Table. Quality assessment of included studies using the appraisal tool for cross-sectional studies (Downes et al., 2016).**
(XLSX)

**S1 Checklist. PRISMA-checklist–transparent reporting of systematic reviews and meta-analyses.**
(DOC)

## Author Contributions

**Conceptualization:** Johanna Lambrich, Thomas Muehlbauer.

**Data curation:** Johanna Lambrich, Thomas Muehlbauer.

**Formal analysis:** Johanna Lambrich, Thomas Muehlbauer.

**Methodology:** Johanna Lambrich, Thomas Muehlbauer.

**Writing – original draft:** Johanna Lambrich, Thomas Muehlbauer.

**Writing – review & editing:** Johanna Lambrich, Thomas Muehlbauer.

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
