## [Decision Letter · Decision Letter 0]

3 May 2022

PONE-D-22-09328Physical fitness and stroke performance in healthy young tennis players with different competition levels: a systematic review and meta-analysisPLOS ONE

Dear Dr. Lambrich,

Thank you for submitting your manuscript to PLOS ONE. After careful consideration, we feel that it has merit but does not fully meet PLOS ONE’s publication criteria as it currently stands. Therefore, we invite you to submit a revised version of the manuscript that addresses the points raised during the review process. In particular, both reviewers have highlighted methodological concerns that must be carefully addressed before accepting your article for publication. 

We look forward to receiving your revised manuscript.

Kind regards,

Javier Peña, Ph.D.

Academic Editor

PLOS ONE

Journal Requirements:

Reviewers' comments:

Reviewer's Responses to Questions

**Comments to the Author**

1. Is the manuscript technically sound, and do the data support the conclusions?

Reviewer #1: Partly

Reviewer #2: Yes

2. Has the statistical analysis been performed appropriately and rigorously? 

Reviewer #1: Yes

Reviewer #2: Yes

3. Have the authors made all data underlying the findings in their manuscript fully available?

Reviewer #1: Yes

Reviewer #2: Yes

4. Is the manuscript presented in an intelligible fashion and written in standard English?

Reviewer #1: Yes

Reviewer #2: Yes

5. Review Comments to the Author

Reviewer #1: I would like to congratulate the authors for this manuscript. In general terms, I consider that the article deals with an extremely interesting topic (i.e., physical fitness and stroke performance in tennis players). Although the paper is well written and structured, in my opinion, it has some methodological aspects that the authors should re-evaluate.

I would like to make some suggestions that should be considered by the authors in order, in my opinion, to improve the quality of the article so that it meets the requirements for publication in the Plos One journal.

Title

The title should not include the term " young" as the study sample includes players up to the age of 32.

Abstract

Write the abstract in a single paragraph, eliminating section headings.

Reduce the length of the abstract, it should not exceed 300 words.

Methodology

Although the authors have made an effort to classify the players into elite and sub-elite, the diversity of levels of the players participating in the different studies makes the heterogeneity of the groups important. But, above all, a major methodological problem is to have included in both groups players in pre-pubescent or pubescent ages (i.e. 12-15) with adult players (i.e. over 18). In this way, it is difficult to justify that the results obtained are due only to the level of the players and not to maturity. The authors should probably include in the analysis only those studies that analyse either young players or adult players. Furthermore, the authors have not made any reference to the age aspect in the discussion.

Line 129: Sub-elite players should not be included in high-performance.

Line 134: Add reference for Dutch Technical-Tactical Test.

Line 137 and/or Table 2: If the most frequent tests were taken as a reference, the number of studies that made use of each test should be indicated in the text and/or in Table 2.

Results

Figure 7: The format of the lines should be changed and a legend should be included indicating which group it represents.

Reviewer #2: Through this manuscript, a systematic review and meta-analysis of physical fitness and stroke performance comparing healthy young elite with sub-elite tennis players is carried out. The greatest differences by competition level were shown in measures of lower extremity muscle power, endurance, and agility, concluding the importance of carrying out training programs for sub-elite tennis players that place special focus on these physical components.

Manuscript is well written and clearly justifies the importance of novelty of the study, especially, considering the importance of ordering the numerous existing scientific literature on stroke performance in healthy tennis players.

However, there are some to improve the document that I indicate below.

The “Abstract” section perfectly synthesizes the different sections of the article.

The “keywords” are correct to facilitate searches, not being redundant with those of the title.

The “Introduction” is complete, precise and progressive to end, in the last paragraph, to describe the aim of the study: "to to additionally quantify differences in physical fitness and stroke performance in healthy young tennis players by competition level ".

Simply, on page 4, line 77, after "and thus for success in a tennis match.", there is a possible bibliographic reference that may be interesting for you, since it is a review that deals with new approaches for on-court endurance testing and conditioning in competitive tennis players, being the aim of this review is to identify a new training load parameter, suitable for on-court use in tennis, based on technical and physiological skills, to allow control of internal and external loads:

Baiget E, Iglesias X, Fuentes JP, Rodriguez FA (2019) New Approaches for On-court Endurance Testing and Conditioning in Competitive Tennis Players. Strength and Conditioning Journal 41: 9-16.

- The "Method" includes all the detailed information and the statistical procedures used are adequate for this systematic review and meta-analysis. Perhaps, it would have been interesting inside "Search strategy" includes not only "forehand OR backhand OR serve" but all those considered by many authors to be 5 basic tennis strokes, where "volley" and "overhead" would be missing. I consider that the number of articles would not have increased much, since there is less scientific literature on "volley" and "overhead" and the systematic review and meta-analysis could be even more complete.

- The “Results” are presented in an orderly manner and the tables and figures are very complete and appropriate.

- The "Discussion" is well structured and supported by adequate bibliographic references.

- The conclusions are well synthetic and indicate the most relevant results of your systematic review and meta-analysis.

6. PLOS authors have the option to publish the peer review history of their article (what does this mean?). If published, this will include your full peer review and any attached files.

Reviewer #1: **Yes: **Rafael Martínez-Gallego

Reviewer #2: No

---

## [Author Response · Author response to Decision Letter 0]

9 May 2022

Reviewer #1: changes were highlighted in yellow

Comment: I would like to congratulate the authors for this manuscript. In general terms, I consider that the article deals with an extremely interesting topic (i.e., physical fitness and stroke performance in tennis players). Although the paper is well written and structured, in my opinion, it has some methodological aspects that the authors should re-evaluate. I would like to make some suggestions that should be considered by the authors in order, in my opinion, to improve the quality of the article so that it meets the requirements for publication in the Plos One journal.

Response: We thank the reviewer for this affirmative comment. All your comments were addressed, and changes were highlighted in yellow.

Comment: Title; The title should not include the term "young" as the study sample includes players up to the age of 32.

Response: In accordance to your suggestion, we deleted the term "young".

Comment: Abstract; Write the abstract in a single paragraph, eliminating section headings. Reduce the length of the abstract, it should not exceed 300 words.

Response: In accordance to your suggestion, we eliminated the section headings and reduced the abstract length to less than 300 words.

Comment: Methodology; Although the authors have made an effort to classify the players into elite and sub-elite, the diversity of levels of the players participating in the different studies makes the heterogeneity of the groups important. But, above all, a major methodological problem is to have included in both groups players in pre-pubescent or pubescent ages (i.e. 12-15) with adult players (i.e. over 18). In this way, it is difficult to justify that the results obtained are due only to the level of the players and not to maturity. The authors should probably include in the analysis only those studies that analyse either young players or adult players. Furthermore, the authors have not made any reference to the age aspect in the discussion.

Response: We thank the reviewer for bringing forward this important issue. Consequently, we performed additional analyses for young (<18 years) and adults ((≥18 years) players, separately. The following changes were made.

Abstract: “However, sub-group analyses revealed an influence of players' age showing higher SMD-values for adult than for young players.”

Methods section (cf. Statistical analysis): “The standardized mean difference (SMD) was used to examine differences due to competition level (i.e., elite vs. sub-elite tennis players) including all player (12-32 years) and for young (<18 years) compared to adult (≥18 years) players [19].”

Results section (cf. Physical fitness differences by competition level): “Further, the age-specific sub-analysis revealed that SMD-values for lower extremity muscle power, endurance, and speed were large in adult but small to moderate in young players (Table 4) that is indicative of a moderating effect of age. For speed, no age-related differences were detected.”

Results section (cf. Stroke performance differences by competition level): “The additionally performed age-specific sub-analysis showed large SMD-values for both the young and the adult players (Table 4), indicating no moderating role of age.”

Table 4. Differences in measures of physical fitness and stroke performance between players with different competition level (i.e., elite vs. sub-elite tennis players) by age group.

Measure Young players

(<18 years) Adult players

(≥18 years) All players

(12-32 years)

Muscle power 0.46 1.12 0.53

Endurance 0.50 0.98 0.59

Agility 0.52 1.02 0.54

Speed 0.35 – 0.35

Stroke performance 0.95 1.15 1.00

Note. Data are presented as standardized mean difference.

Discussion section (cf. Differences in physical fitness by competition level): “In addition, the size of the SMD-values differed by age group for measures of lower extremity muscle power, endurance, and agility. Specifically, differences between elite and sub-elite players in these physical fitness components were greater for adult than for young players. This implies that with increasing age it becomes more important to train lower extremity muscle power, endurance, and agility in order to perform successful. In this regard, Kurtz et al. reported a significant correlation between national ranking and agility in adult tennis players. In contrast, no significant correlations between national ranking and lower extremity muscle power were detected for young players (Colomar, Ulbricht).

Discussion section (cf. Differences in stroke performance by competition level): “Further, the size of the SMD-values did not differ by age group and was large in young and adult players as well. This suggests that it is significant to practice tennis specific skills already at a young age. In fact, it is recommended in the guidelines of the German Tennis Confederation to practice stroke techniques especially in young players (i.e., from under 10 to under 14 years).”

Line 129: Sub-elite players should not be included in high-performance.

Response: We apologize for this typo and deleted the term "high-performance" from the classification of sub-elite players.

Line 134: Add reference for Dutch Technical-Tactical Test.

Response: The following reference for the Dutch Technical-Tactical Test was added as suggested: “Kolman N, Huijgen B, Kramer T, Elferink-Gemser M, Visscher C. The Dutch Technical-Tactical Tennis Test (D4T) for Talent Identification and Development: Psychometric Characteristics. Journal of human kinetics 2017; 55: 127–38 [https://doi.org/10.1515/hukin-2017-0012][PMID: 28210345]”

Line 137 and/or Table 2: If the most frequent tests were taken as a reference, the number of studies that made use of each test should be indicated in the text and/or in Table 2.

Response: We agree with the reviewer and added the number of studies that made use of each test to Table 2.

Comment: Results; Figure 7: The format of the lines should be changed and a legend should be included indicating which group it represents.

Response: We thank the reviewer for this note and changed the format of the regression lines. Specifically, the solid regression line indicates elite tennis players and the dotted regression line means sub-elite tennis players. This is now stated in the Figure legend.

 

Reviewer #2: changes were highlighted in green

Comment: Through this manuscript, a systematic review and meta-analysis of physical fitness and stroke performance comparing healthy young elite with sub-elite tennis players is carried out. The greatest differences by competition level were shown in measures of lower extremity muscle power, endurance, and agility, concluding the importance of carrying out training programs for sub-elite tennis players that place special focus on these physical components. Manuscript is well written and clearly justifies the importance of novelty of the study, especially, considering the importance of ordering the numerous existing scientific literature on stroke performance in healthy tennis players. However, there are some to improve the document that I indicate below.

Response: We thank the reviewer for this affirmative comment. All your comments were addressed, and changes were highlighted in green.

Comment: The “Abstract” section perfectly synthesizes the different sections of the article.

Response: Thank you for this affirmative comment.

Comment: The “keywords” are correct to facilitate searches, not being redundant with those of the title.

Response: We thank the reviewer for this confirmative comment.

Comment: The “Introduction” is complete, precise and progressive to end, in the last paragraph, to describe the aim of the study: "to to additionally quantify differences in physical fitness and stroke performance in healthy young tennis players by competition level ".

Response: The superfluous "to" was deleted.

Comment: Simply, on page 4, line 77, after "and thus for success in a tennis match.", there is a possible bibliographic reference that may be interesting for you, since it is a review that deals with new approaches for on-court endurance testing and conditioning in competitive tennis players, being the aim of this review is to identify a new training load parameter, suitable for on-court use in tennis, based on technical and physiological skills, to allow control of internal and external loads:

Baiget E, Iglesias X, Fuentes JP, Rodriguez FA (2019) New Approaches for On-court Endurance Testing and Conditioning in Competitive Tennis Players. Strength and Conditioning Journal 41: 9-16.

Response: We thank the reviewer for this comment and agree to add the stated references.

Comment: The "Method" includes all the detailed information and the statistical procedures used are adequate for this systematic review and meta-analysis. Perhaps, it would have been interesting inside "Search strategy" includes not only "forehand OR backhand OR serve" but all those considered by many authors to be 5 basic tennis strokes, where "volley" and "overhead" would be missing. I consider that the number of articles would not have increased much, since there is less scientific literature on "volley" and "overhead" and the systematic review and meta-analysis could be even more complete.

Response: We thank the reviewer for this legitimate comment. In accordance to your suggestion, we added the "volley" and "overhead" to the search term. As a consequence, we obtained 778 more articles, but they were not included in the final analyses for the following reasons: 25 records were identified as duplicates, 715 records were excluded based on title, and 38 records were excluded based on abstract. However, the following changes were made.

Abstract: “A systematic literature search was conducted in the databases PubMed, Web of Science, and SportDiscus from their inception date till May 2022.” and “The search identified a total of N = 12,714 records, 16 of which met the inclusion criteria.”

Methods section (cf. Search strategy): “The following Boolean search term was used: (tennis AND ((performance level OR competition level OR elite OR expert OR high performance OR non-expert OR sub-elite OR amateur NOR Novice NOR beginner) OR (stroke OR physical fitness OR fitness characteristics OR agility OR endurance OR speed OR muscle power OR lower extremity NOT upper extremity) OR (forehand OR backhand OR serve OR volley OR overhead)) NOT table).”

Results section (cf. Study selection): “The search identified a total of N = 12,714 records, 16 of which met the inclusion criteria.”

Comment: The “Results” are presented in an orderly manner and the tables and figures are very complete and appropriate.

Response: We thank the reviewer for this confirmative comment.

Comment: The "Discussion" is well structured and supported by adequate bibliographic references.

Response: Thank you for agreeing with the content and structure of the Discussion section.

Comment: The conclusions are well synthetic and indicate the most relevant results of your systematic review and meta-analysis.

Response: Thank you very much for your affirmative comment.

References used for revision:

Baiget E, Iglesias X, Fuentes JP, Rodriguez FA (2019) New approaches for on-court endurance testing and conditioning in competitive tennis players. Strength and Conditioning Journal 41: 9-16.

Colomar, Joshua; Baiget, Ernest; Corbi, Francisco (2020): Influence of Strength, Power, and Muscular Stiffness on Stroke Velocity in Junior Tennis Players. In: Frontiers in Physiology 11, S. 196.

Eberhard, K.; Fratzke, G.; Jansen, E.; Januschke, J.; Krelle, K.; Spreckels, C. (2019): Rahmentrainingskonzeption des Deutscher Tennis Bund e.V. Training methodological framework of the German Tennis Federation. Online verfügbar unter https://www.dtb-tennis.de/content/download/19819/205357/version/1/file/Rahmentrainingskonzeption.pdf.

Kolman N, Huijgen B, Kramer T, Elferink-Gemser M, Visscher C. The Dutch Technical-Tactical Tennis Test (D4T) for talent identification and development: psychometric characteristics. Journal of Human Kinetics 2017; 55: 127–38.

Kurtz, J. A.; Grazer, J.; Alban, B.; Marino, M. (2019): Ability for tennis specific variables and agility for determining the Universal Tennis Ranking (UTR). In: The Sports Journal. 

Ulbricht, Alexander; Fernandez-Fernandez, Jaime; Mendez-Villanueva, Alberto; Ferrauti, Alexander (2016): Impact of fitness characteristics on tennis performance in elite junior tennis players. In: Journal of strength and conditioning research 30 (4), S. 989–998.

---

## [Decision Letter · Decision Letter 1]

24 May 2022

Physical fitness and stroke performance in healthy tennis players with different competition levels: a systematic review and meta-analysis

PONE-D-22-09328R1

Dear Dr. Lambrich,

We’re pleased to inform you that your manuscript has been judged scientifically suitable for publication and will be formally accepted for publication once it meets all outstanding technical requirements.

Kind regards,

Javier Peña, Ph.D.

Academic Editor

PLOS ONE

Additional Editor Comments (optional):

Reviewers' comments:

Reviewer's Responses to Questions

**Comments to the Author**

1. If the authors have adequately addressed your comments raised in a previous round of review and you feel that this manuscript is now acceptable for publication, you may indicate that here to bypass the “Comments to the Author” section, enter your conflict of interest statement in the “Confidential to Editor” section, and submit your "Accept" recommendation.

Reviewer #1: All comments have been addressed

Reviewer #2: All comments have been addressed

2. Is the manuscript technically sound, and do the data support the conclusions?

Reviewer #1: (No Response)

Reviewer #2: Yes

3. Has the statistical analysis been performed appropriately and rigorously? 

Reviewer #1: (No Response)

Reviewer #2: Yes

4. Have the authors made all data underlying the findings in their manuscript fully available?

Reviewer #1: (No Response)

Reviewer #2: Yes

5. Is the manuscript presented in an intelligible fashion and written in standard English?

Reviewer #1: (No Response)

Reviewer #2: Yes

6. Review Comments to the Author

Reviewer #1: (No Response)

Reviewer #2: The authors have implemented all the suggestions of this reviewer.

I congratulate the authors for the good work they have done, which has substantially improved the initial manuscript.

As I indicated in the first review, the manuscript clearly justifies the importance of novelty of the study, especially, considering the importance of ordering the numerous existing scientific literature on stroke performance in healthy tennis players.

7. PLOS authors have the option to publish the peer review history of their article (what does this mean?). If published, this will include your full peer review and any attached files.

Reviewer #1: **Yes: **Rafael Martínez-Gallego

Reviewer #2: No